# Topical Cellular/Tissue and Molecular Aspects Regarding Nonpharmacological Interventions in Alzheimer’s Disease—A Systematic Review

**DOI:** 10.3390/ijms242216533

**Published:** 2023-11-20

**Authors:** Sorina Aurelian, Adela Ciobanu, Roxana Cărare, Simona-Isabelle Stoica, Aurelian Anghelescu, Vlad Ciobanu, Gelu Onose, Constantin Munteanu, Cristina Popescu, Ioana Andone, Aura Spînu, Carmen Firan, Ioana Simona Cazacu, Andreea-Iulia Trandafir, Mihai Băilă, Ruxandra-Luciana Postoiu, Andreea Zamfirescu

**Affiliations:** 1Faculty of Medicine, University of Medicine and Pharmacy “Carol Davila”, 020022 Bucharest, Romania; sorina.aurelian@umfcd.ro (S.A.); adela.ciobanu@umfcd.ro (A.C.); cristina_popescu_recuperare@yahoo.com (C.P.); ioanaandone11@yahoo.com (I.A.); aura_ko@yahoo.com (A.S.); andreea-iulia.trandafir@drd.umfcd.ro (A.-I.T.); mihai.baila@rez.umfcd.ro (M.B.); postoiu.ruxandra@yahoo.ro (R.-L.P.); andreea.zamfirescu@umfcd.ro (A.Z.); 2Gerontology and Geriatrics Clinic Division, St. Luca Hospital for Chronic Illnesses, 041915 Bucharest, Romania; 3Department of Psychiatry, ‘Prof. Dr. Alexandru Obregia’ Clinical Hospital of Psychiatry, 041914 Bucharest, Romania; 4Faculty of Medicine, University of Southampton, Southampton SO16 7NS, UK; r.o.carare@soton.ac.uk; 5NeuroRehabilitation Clinic Division, Teaching Emergency Hospital “Bagdasar-Arseni”, 041915 Bucharest, Romania; stoica.simona@umfcd.ro (S.-I.S.); aurelian.anghelescu@umfcd.ro (A.A.); ioanas.cazacu@gmail.com (I.S.C.); 6Faculty of Midwifery and Nursing, University of Medicine and Pharmacy “Carol Davila”, 020022 Bucharest, Romania; 7Computer Science Department, Politehnica University of Bucharest, 060042 Bucharest, Romania; vlad.ciobanu@upb.ro; 8Faculty of Medical Bioengineering, University of Medicine and Pharmacy “Grigore T. Popa”, 700115 Iași, Romania; 9NeuroRehabilitation Compartment, The Physical and Rehabilitation Medicine & Balneology Clinic Division, Teaching Emergency Hospital of the Ilfov County, 022104 Bucharest, Romania; firancarmen@yahoo.com

**Keywords:** Alzheimer’s disease, amyloid-beta (Aβ) aggregation, tau hyperphosphorylation, neuroinflammation, nonpharmacological interventions, neuroplasticity

## Abstract

One of the most complex and challenging developments at the beginning of the third millennium is the alarming increase in demographic aging, mainly—but not exclusively—affecting developed countries. This reality results in one of the harsh medical, social, and economic consequences: the continuously increasing number of people with dementia, including Alzheimer’s disease (AD), which accounts for up to 80% of all such types of pathology. Its large and progressive disabling potential, which eventually leads to death, therefore represents an important public health matter, especially because there is no known cure for this disease. Consequently, periodic reappraisals of different therapeutic possibilities are necessary. For this purpose, we conducted this systematic literature review investigating nonpharmacological interventions for AD, including their currently known cellular and molecular action bases. This endeavor was based on the PRISMA method, by which we selected 116 eligible articles published during the last year. Because of the unfortunate lack of effective treatments for AD, it is necessary to enhance efforts toward identifying and improving various therapeutic and rehabilitative approaches, as well as related prophylactic measures.

## 1. Introduction

Currently, the population faces one of humanity’s greatest and most multifaceted global challenges—an almost worldwide demographic aging process [1,2]. Consequently, the number of people living with dementia comprises over 55 million people (World Health Organization. (2021). Dementia fact sheet. https://www.who.int/news-room/fact-sheets/detail/dementia–last accessed on 28 August 2023). Alzheimer’s disease (AD)—named after Dr. Alois Alzheimer—represents up to 80% of all dementia cases and accounts for over 6 million persons affected in the USA [3,4]. Thus, AD is “the most/main common form/type/cause of dementia“ [3,5,6,7,8] and the “most widely studied” type [9], followed by other types of dementia, such as vascular dementia with Lewy bodies or frontotemporal [10] and mixed dementia. If AD is considered in the prodromal/preclinical AD stage, it results in “416 million across the AD continuum” [11]. AD occurs across all continents and among all races, with the incidence being greater in women [12]. Aging is the most important related risk factor^2^, and as such, most patients are elderly, i.e., ≥65 years old^2^ [13]. Sporadic AD affects those over the age of 65, and its incidence increases with age. Familial AD has a lower age of onset and is rare in comparison to the sporadic form [14]. It is also specified that sporadic AD has, as a major risk factor, a higher prevalence of the ε4 allele of apolipoprotein E (ApoE), which is also the case in late-onset familial AD [15,16]. AD, as an expression/consequence of a non-physiological way of growing old, progressively impairs basic cognition, including memory and the habits that rely on it, as well as somatic functions. Affected individuals with disabled status eventually see the disease compromise their activities of daily living^2^; at the same time, AD is the fifth main cause of mortality in the elderly (≥65 years old) [17]. Cognitive decline is determined by an increased rate of neuronal death/losses in the brain, through complex and still incompletely understood causes [17]. It may also entail chronic pain [18] and motricity and mobility impairments, especially in advanced stages; movements, including walking, are progressively slowed, with rigid muscle hypertonia of the extrapyramidal kind. Hence, reduced diversity in activities may predict AD onset [19]. In addition to the above-mentioned symptoms, there are language processing issues [20] and problem-solving difficulties [10] that accompany the progressive deterioration of episodic and semantic memory, language, and visuospatial ability [14], all within a taxonomic cluster referred to as behavioral and psychological symptoms of dementia (BPSD) [21]. These are also referred to as neuropsychiatric symptoms of dementia and include changes in behavior, perception, content of thoughts, and mood disorders that are found in AD as well as other types of dementia [5].

Therefore, dementia, including its most common subtype, AD, represents a growing “serious global public health problem”, with multifaceted consequences and challenges that affect individuals, their kin, and the wider community, and have broad socioeconomic impacts [22]. For example, annually in the USA, the overall financial burden of related therapeutic approaches approximates USD 305 billion, and this cost may soon grow, exceeding USD 1 trillion, because of inevitable demographic aging [23].

Current treatments, most of which are medicine-based, have little or no effect on the evolution/progression of this disease. Accordingly, the scope of our systematic literature review is to highlight the basic mechanisms behind the development of AD in connection with the role of its risk and diagnosis factors, and to use this information for a comprehensive topical reappraisal of the most recent literature concerning nonpharmacologic therapeutic and rehabilitative interventions, along with their intimate, cellular, and molecular actions, thus addressing the gap in the structured knowledge in this domain.

Regarding AD’s pathogeny, one of the major causes is the age-related failure of the elimination of a small protein, amyloid-β (Aβ), that results in its accumulation in the brain and the walls of arteries supplying the brain [24]. Soluble waste substances, including Aβ, drain from the brain along thin membranes in the walls of capillaries and arteries in a process known as Intramural Periarterial Drainage (IPAD). With advancing age, in AD, Aβ is deposited within these membranes, further impeding the elimination of Aβ and other waste material from the brain [25,26]. It should be emphasized that when predicting the appearance of AD, the impaired purging of the Aβ precursor protein is considered a major inceptive disturbance at the intimate level and results in the pathogenic Aβ peptide’s “accumulation and plaque formation 20 to 30 years before cognitive symptoms arise” [27]. Apart from age, other risk factors for AD may also impede the elimination of Aβ from the brain. Such risk factors are present in patients who produce the epsilon 4 (ε4) form of the protein apolipoprotein E (apoE4), particularly in those with high levels of cholesterol in the blood [28,29] and/or hypertension and/or with “metal-induced neurotoxicity” (iron, copper, and manganese are particularly important elements required for brain function and development); their local imbalance can be functionally detrimental, i.e., neurodegenerative, at the cerebral level, more precisely affecting “the motor, cognitive and emotional systems” [30]. Further consideration of, on the one hand, the complex path-physiologic mechanisms of AD at the intimate level and, on the other hand, the abovementioned necessity to detect AD as early as possible at the presymptomatic stage has led to the recent expansion of the focus on para-clinical diagnosis. Alongside a blood assay to check for “blood count, thyroxine, antithyroid antibodies, anti-syphilis antibodies, folic acid, and vitamin B12”, positron emission tomography (PET) is quite a valuable nuclear medicine tool that is able to investigate the prodromal and advanced dementia stages. To examine isotopic markers, AD-related PET examinations can be used, namely, Aβ-PET, tau-PET, and fluorodeoxyglucose (FDG)-PET [31]. As supplementary proof of its very complex and still incompletely understood causality, there is an example that does not only apply to older people: post-traumatic encephalopathy (PTE) may evolve into secondary Alzheimer-like dementia even decades after the trauma [32]. Moreover, the literature explicitly emphasizes a social–biopathological link too. Loneliness can be mentally distressing and can also induce autonomic and phlogistic responses [33]. This is considered to occur in AD as a consequence of the inflammation caused by the enhancement of oxygen toxicity-free-radical production. Oxidative stress [34,35], which produces “oxidation of the chemicals induced by free radicals” [36], causes damage when reactive oxygen species (ROS) reach an augmented amount [37]. They have especially aggressive actions in the hippocampus, which is part of the brain structure involved in memory and executive functions. So, in terms of immune–endocrine functional interference, loneliness may be considered in relation to a subsequent lingering hyperactivity of the hypothalamus–pituitary–adrenal (HPA) axis [33]. In fact, generally, aging also results in a “progressive chronic pro-inflammatory state”, and this is considered a basic biological issue (together with “the critical role of a dysregulated immune system in promoting persistent neuroinflammation”) [38], encountered in different age-related sicknesses, including AD [39] and Parkinson’s disease (PD) [38].

### 1.1. Pathology of AD

The pathological features of AD are neurofibrillary tangles, amyloid plaques, and vascular amyloidosis [40]. Neurofibrillary tangles are non-membrane-bound bundles of paired helically wound filaments, 6–11 nm in width and 10 nm–2 µm in length, formed from the microtubule-associated protein tau in its insoluble form. When stained with Congo red, neurofibrillary tangles possess green birefringence in polarized light [41]. The insoluble tau within the filaments is covalently bound to ubiquitin [42].

In the widely accepted amyloid cascade hypothesis for the pathogenesis of AD, the Aβ protein is considered to be the single key element responsible for neurodegeneration and dementia through a series of related events [43]. ApoE, extracellular matrix, and basement membrane components have been located in AD plaques [44]. The Aβ peptide accumulates in its insoluble form in the walls of cerebral blood vessels as cerebral amyloid angiopathy (CAA) [45].

The diagnostic criteria for AD are grouped into a clinical and neuropsychological assessment within the consortium to establish a registry for Alzheimer’s disease and pathological staging according to the neurodegenerative features [46,47]. The deposition of Aβ in blood vessel walls as CAA correlates strongly with the presence of dementia and is now considered a key feature in diagnosing AD [48].

Aβ is produced systemically by most tissues [49]. There is evidence for the uptake of Aβ from blood and transport across the blood-brain barrier [50]. In earlier studies, smooth muscle cells were named as the sole source of Aβ in vessel walls; however, this does not provide an explanation as to why Aβ does not accumulate in the walls of smooth-muscle-rich extracranial vessels [51]. More recently, it has been shown that Aβ is produced by neurons [52]. The overexpression of the mutant human amyloid precursor protein in the neurons of transgenic mice induced amyloid deposition in blood vessel walls and associated neurodegeneration [53]. Thus, in transgenic mice, the transport and drainage of Aβ from neurons is responsible for the deposition of Aβ in blood vessel walls [53]. Soluble Aβ is predominantly 1–40 (Aβ40) and is found at low concentrations in normal brains [54]. The levels of soluble Aβ40 and 42 are in the picomolar range in young normal brains and increase 100–1000-fold with age and in CAA [55]. The amyloid in neuritic plaques consists of 42–43 amino acids at its C-terminal [55]. Vascular amyloid mainly consists of the 1–39/40 type, with some Aβ 1–42 [55].

Aβ is derived through cleavage from a transmembrane-secreted protein: Amyloid Precursor Protein (APP). The gene encoding of APP is located on chromosome 21 [56]. APP is involved in homeostasis, regulation, and neuroprotection. Cleavage by β and γ secretases results in the generation of Aβ, whereas γ secretase cleaves APP within the Aβ sequence, thus preventing the formation of Aβ [57].

ApoE is secreted by neurons under the influence of regulatory astrocytic-secreted factors [58]. The three common alleles ε2, ε3, and ε4 encode the E2, E3, and E4 isoforms of ApoE. ApoE has been described as a chaperone molecule for Aβ [58]. Evidence from human and transgenic mouse studies shows that the ε4 isoform of ApoE (ApoE4) is a predisposing factor for AD, hypercholesterolemia, atherosclerosis, and poor outcomes after head injuries [59,60]. Epidemiological studies estimate that approximately one-third of the population are ApoE4 heterozygotes; this is associated with a two- to threefold increased risk of developing AD. The 1–2% of the population who are ApoE4 homozygotes have an approximately 8–10-fold risk of AD. The mechanism by which ApoE exerts its effects on Aβ transport and deposition is unclear, but it seems to be related to a worsening of the clearance of Aβ [28,61]. On the other hand, clusterin (apolipoprotein J) appears to be involved in blocking the aggregation of Aβ and preventing its deposition [62].

### 1.2. Familial and Sporadic AD

Familial forms of AD appear to be due to an overproduction of Aβ as a result of mutations or polymorphisms in one of three genes. Mutations in the APP (amyloid precursor protein) gene account for less than 0.1% of all AD cases but form the basis for one of the most informative transgenic mouse models of AD [63]. Presenilin-1 (PS1) and presenilin-2 (PS2) span cellular membranes and are encoded on chromosomes 14 and 1, respectively [64]. More than 50 mutations have been described for PS1, compared to eight in PS2 [65]. Double transgenic mice for APP and PS1 express more Aβ42 and develop a larger number of Aβ deposits in their brains than single APP transgenic mice. Although Aβ deposits are not present, there is an increase in the concentration of Aβ42 in PS1 single transgenic mice, suggesting that presenilin mutations enhance the secretion of the more amyloidogenic Aβ42 [66].

In the familial forms of cerebral amyloid angiopathy, there are other amyloid proteins that deposit in blood vessel walls: the mutant cystatin C in hereditary cerebral hemorrhage with amyloidosis, the Icelandic-type variant transthyretins in meningo-vascular amyloidosis, mutant gelsolin in Finnish-type familial CAA, PrPSc in Creutzfeldt–Jakob disease and a variant of Gerstmann–Straüssler–Scheinker syndrome, ABri in familial British dementia, and ADan in familial Danish dementia [67].

The great majority of cases of AD are sporadic, and there appears to be no firm evidence of the overproduction of Aβ. It seems, therefore, that the failure to eliminate Aβ from the elderly brain is an important factor in the pathogenesis of sporadic AD. The pattern of distribution of Aβ in the blood vessel walls in both human and mouse CAA suggests that Aβ is eliminated from the brain along Intramural Periarterial Drainage (IPAD) pathways that fail with advancing age and the possession of the APOE4 genotype [26,68]. Other hypotheses for the pathogenesis of sporadic AD include, but are not limited to, the role of microbial infections via periodontal and gastrointestinal routes, as systemic inflammation can drive the amyloid cascade in the brain [69,70]. Age-related oxidative stress leads to mitochondrial dysfunction, which can then trigger a cascade leading to the aggregation of Aβ and the intraneuronal accumulation of tau [71].

## 2. Methods

In order to carry out a systematic literature review on this subject, we followed the principles of the Preferred Reporting Items for Systematic Reviews and Meta-Analyses (PRISMA), with preliminary registration of it on the International Prospective Register of Systematic Reviews [72] (PROSPERO) website (related ID: 456157) (https://www.crd.york.ac.uk/PROSPERO, last accessed on 28 August 2023). For this purpose, we followed the five-step filter/selection PRISMA-specific standardized methodology. First, we contextually interrogated four renowned international medical databases: Elsevier (available at https://www.elsevier.com/ (last accessed on 28 August 2023)), the National Center for Biotechnology Information (NCBI)/PubMed (available at https://pubmed.ncbi.nlm.nih.gov/ (last accessed on 28 August 2023)), the National Center for Biotechnology Information (NCBI)/PubMed Central (PMC) (available at https://www.ncbi.nlm.nih.gov/pmc/ (last accessed on 28 August 2023)), and the Physiotherapy Evidence Database PEDro (available at https://www.strokengine.ca/glossary/pedro--score (last accessed on 28 August 2023)), using the keyword combinations/syntaxes displayed in Table 1. The selection criteria for the studies considered in our work were free full-text papers, written in English, and published between 1 January and 31 December 2022 in journals indexed in the Institute for Scientific Information (ISI—ex Thomson Reuters—currently administered by Clarivate Analytics), i.e., the renowned ISI Web of Knowledge/Science (The Institute for Scientific Information, Web of Science Group. Available at https://mjl.clarivate.com/home (last accessed on 28 August 2023)) database.

In the next step, we indirectly assessed the scientific quality of the remaining articles (after traversing the previous steps) using a proper related measurement-weighted algorithm [73] inspired by the PEDro classification/scoring system, selecting papers that obtained a score of at least 4 (“fair quality = PEDro score 4–5”) (PEDro score, Strok Engine. Available at strokengine.ca/glossary/pedro--score/ (last accessed on 28 August 2023)).

In another step, we excluded papers that appeared to be eligible according to the criteria mentioned, which after reading, were found to be irrelevant/non-contributive in relation to our scope (“full-text articles excluded with reasons”). This was based on a direct quality assessment regarding the relevance of the papers’ content to our focus. As for the data extraction from the selected articles, each author was allotted a number of papers for condensed evaluation and preliminary data synthesis. In the last step, we analyzed all of the papers deemed to contain the most useful information and extracted the data underpinning the work (Figure 1). A full list of references and links to the final selected works within our systematic literature review are presented in Section 3.

We acknowledge that, despite the rigor of the PRISMA method, it is still possible that some papers that would have been useful to this project may have been overlooked. We therefore added a number of contributive papers freely found in the literature to the knowledge base of this systematic review [74].

## 3. Results

Generally, and specifically in this paper, nonpharmacological interventions refer to medical, psychological, and social therapeutic–rehabilitative endeavors (see Table 2) without the use of medicines. Consequently, they are very diverse and therefore difficult to classify, encompassing cognitive therapy and physical exercise either alone or within multimodal procedures [75]. An appropriate attempt to systematize these approaches to the treatment of BPSDs (but without encompassing all of their diversity, e.g., physiatric types of interventions, which will be approached differently, or dose methodology-dependent outcomes) entails directed interventions such as “reminiscence therapy, validation therapy, and supportive psychotherapy”, “reality orientation and skills training”, and “recreational activities, art therapies, exercise, and music therapies”. Yet, we believe this structure does not sufficiently encompass all of their diversity, for instance, physiatric types of interventions [21]. So, as nonpharmacological interventions, including for the treatment of AD, are quite diverse and at least some of them are either mixed and/or rather nondescript, we opted to arrange them in two complementary ways: “classical”, i.e., grouped mainly as presented below, and, respectively, in tabular form according to their names, that is, in alphabetic order.

The heterogeneity of this diverse field matched with the consensus criteria constructed in the Delphi technique (DT) [76], with this being the main reason for the use of a muti-professional panel (physicians: Gerontology and Geriatrics, Physical and Rehabilitation Medicine, Psychiatry, Neurology, Neuropathology, Neurobiology, and IT).

It is also worth noting that according to objective editorial customs, this paper must be of a reasonable length; therefore, herein, “we shall not detail related methodological aspects” [77].

### 3.1. Acupuncture

Some of the literature data regarding related trials report that this type of intervention, targeting specific acupoints, can support the flow of energy across the acupuncture meridians (https://acupuncturecanada.org/acupuncture-101/what-is-acupuncture/ ((last accessed on 28 August 2023)) to “replenish qi resolve phlegm, and promote blood circulation”, outcomes objectified through favorably modified scores on the Alzheimer’s Disease Assessment Scale—cognitive subscale (ADAS-cog) and Clinician’s Interview-Based Impression of Change—Plus (CIBIC-Plus) of AD patients [78]. Interestingly, studies in this field “using data mining methods” provided algorithms for appropriate acupoints that match different types of neurological pathology, including AD [79]. Specifically, “in the treatment of AD with acupuncture and moxibustion”, many acupoints are “selected from the Governor Vessel”, being prevalent in the “combination of the local acupoints with the distal ones” [80].

#### Electroacupuncture

This kind of intervention is a newer form of acupuncture that uses low-intensity electric currents delivered through small electrodes connected to acupuncture needles inserted into specific acupoints to augment the therapeutic effects (https://www.webmd.com/pain-management/what-is-electroacupuncture ((Last accessed on 28 October 2023)); applied at “at acupoints on the head”, these have shown outcomes quantified by the enhancements in the scores on the Montreal Cognitive Assessment (MoCA), in patients with AD [78].

### 3.2. Cognitive Behavioral Therapy (CBT)

The method, based on first determining abnormal cognition, emotions, and or habitus reactions, “is a type of psychotherapeutic treatment that… combines cognitive therapy with behavior therapy”, thus improving the affected individual’s ability to cope (Kendra Cherry (Updated on 10 August 2022). Medically reviewed by Rachel Goldman–https://www.verywellmind.com/what-is-cognitive-behavior-therapy-2795747, accessed on 28 October 2023). This nonpharmacological type of intervention seems to be useful in the treatment of insomnia. Such chronically affected persons appear to be prone to cognitive decline earlier in life and, therefore, to developing AD and “Usual CBT (CBT-I–cognitive behavioral therapy for insomnia–o. n.)… enhance cognitive function”; moreover, at the intimate level, the “preliminary findings suggest that CBT may reduce the rate of Aβ deposition in older adults with insomnia and potentially delay” the onset of AD [81].

#### Counseling and Psychoeducation

This kind of procedure is quite complementary, possibly merging with CBT, as it “provides systematic disease-specific information”, contributing to the promotion of a healthy lifestyle, including better coping strategies [82]. Such types of interventions (including “multifaceted and semi-tailored counseling, education, and support” [83]), administered for a longer time, with telephone tracking and related guidance provision [84], showed a “small positive effect” in the treatment of depression in patients with mild AD, according to the Danish Alzheimer Intervention Study (DAISY) [83]. As the implications, at the intimate level, of different environmental destressing agents are well known (for instance, the influence upon the expression of the brain-derived neurotrophic factor (BDNF) “in specific brain regions” [85]), we do not discuss these aspects in any further detail.

### 3.3. Environmental Adjustment

Connected to this subject matter, “loneliness and social isolation”—with marked detrimental psychological and biological consequences, including favoring the development of AD—are, as already pointed out from different perspectives, a matter of public health and policy [86]. So, combating these issues has become an important social/community goal “in a demographically aging population” [87]. Moreover, a clinical (the University of California (UCLA) Los Angeles Loneliness Scale [88], respectively revised [89]) and imagistic (i.e., magnetic resonance imaging (MRI)) evaluation instrument has been developed to objectify and assess the consequences of loneliness: “the loneliness score was significantly negatively correlated with rWMD” (regional WM density) “in eight clusters” of the brain cortex [90]. Being rather eclectic and with a variable taxonomical framing, an approach found in the literature designed to mitigate loneliness in the elderly was the physical practice of “reminiscence therapy, and technological interventions”, used as a community contribution to “improve the social milieu of older adults” [91]. Related examples include “community-based services (i.e., meeting centers, Alzheimer’s Cafés)” [92] and/or “organizing social events” [87]. Even deeper within the intimacy of the superior nervous activity’s biological support are “Musical Abilities, Pleiotropy (genetic kind), Language, and Environment”, including a “musically and linguistically enriched” (MAPLE) integrative framework. This is a complex endeavor that may be helpful to add necessary knowledge about the alterations to underlying premonitory semiology aspects—and even maybe related biomarkers—of communication problems [93]. Additional related endeavors include “Modification to the built environment, Fall prevention, Digital Health” [94]. Another dimension of environmental adjustment is oriented towards making it more agreeable. In this respect, aromatherapy is a common procedure used among adults, including healthy individuals, as a wellness, relaxation, and non-standardized procedure. As medicines based on acetylcholinesterase inhibitors are the ones used more often in AD, and considering, in this respect, the possible helpful composition of Salvia officinalis [95], the use of the “essential oil (EO) of Salvia officinalis (common sage)” [96] was recommended in the literature for the treatment of AD [97].

### 3.4. Exercise/Physical Activity

These kinds of “low-cost accessible” [98] nonpharmacological procedures are frequently discussed in many studies, as they have attracted growing interest in recent decades. This results from the accumulation of enhanced and deeper knowledge in most of the medical domains, and in this context, consistent with evidence-based principles, the benefits of physical activity are now more well-established. The benefits, as well as the limits and even adverse effects, of a larger amount of pharmacological and nonpharmacological interventions, are now known. Therefore, the literature provides a topical, balanced, and more holistic approach paradigm that we consider largely applicable. In terms of specific, disease-centered treatment, “a multidisciplinary support intervention program should contain pain therapy, nutritional medicine, and exercise therapy” [99]. Generally, in addition to its ability to counteract depression, including its onset, there are known beneficial actions of physical activity, especially when integrated within a healthy pro-active lifestyle. Improvements to body image and self-esteem, as well as self-capabilities to efficiently solve daily tasks, consequently improve quality of life (QOL), thus contributing to the reduction in the occurrence or development of cognitive problems [100]. Overall, a proactive lifestyle, including controlled physical exercise or sports, could mitigate the risk of AD appearance and postpone “the onset of loss of autonomy by 7 to 10 years” [101]. Such interventions produce extensive effects. Hence, physical exercise can adaptatively promote the general homeostatic balance of the organism, and at the same time, affect biological states and functions, from the intimate level of cells and tissues to organs, apparatuses, and systems. For instance, “a single exercise session is sufficient to produce acute changes at the transcriptional level”, and, if repeatedly practiced, through related adaptation, exercise can generate “more lasting effects on protein function” [102]. In elderly humans, exercise enhances memory and learning and slows down mental decline. The onset of AD is delayed in individuals who exercise, and there is some evidence that the cognitive decline may be delayed [103], including the paraphysiological aspects associated with aging and also the onset of dementias (e.g., AD), based on complex, subtle, multi-target actions at various morpho-physiological levels [96,97,98]. Exercise may be most beneficial in individuals who are ApoE4-positive, although this requires further study. Recent imaging studies in young humans demonstrate that 12 weeks of exercise aimed at cardiovascular benefit increases blood flow in the hippocampus, and this is associated with improved learning tasks. Taken together, these observations suggest that exercise has a beneficial effect on the function of the cerebral vasculature, resulting in a more efficient clearing of the toxic soluble Aβ from the aging brain [104]. Yet, although this type of procedure could augment the blood flow in the brain (as in the case of AD), the direct effective action of this augmentation on intellective functions is questionable [98]. However, the regional cerebral blood flow (rCBF) is “an index of prefrontal regional brain oxygenation”, and its fluctuations are considered detrimental to cognition, thereby possibly also reflecting the recovery process in traumatic brain injury (TBI), AD, and mild cognitive decline (MCI). Even just walking at ”low or moderate intensity” may ameliorate “cognitive function mediated by the increased prefrontal oxygenation” it produces [105]. Additionally, exercise enhances brain metabolic activity and the synthesis/release of BDNF, “which support brain plasticity and angiogenesis in the hippocampus” [106]. As the amount of BDNF being disturbed in AD is high [107], the ability of physical exercise to increase BDNF is widely pointed out in the literature [108,109]. Regarding the efficacy of exercise in terms of brain biology and the modulation of BDNF levels, resistance training (RT) “at moderate intensity” is recommended and has proven effective. Accounting for individual tolerance is recommended but ought to be in accordance with patient-centered needs, possibilities, and choices in order to ensure adherence to an exercise regimen [110]. Further exploration of the related actions at the intimate biological level revealed that exercise “can influence the transcriptome characteristics of monocytes”. Actually, at the intimate level, moderate exercise seems to favor the molecular expression associated with oxidized low-density lipoprotein (LDL) of induced trans-endothelial monocytes’ passage, including their adhesion. Based on these stimulated actions, monocytes have a potential role in the control and prevention of AD [111]. Additionally, Irisin, which is a myokine largely expressed in different tissues and organs, decreases with age and, hence, is possibly involved in different illnesses of this type. Thus, it may contribute to the amelioration of such diseases, including AD. The main intimate mechanisms involved seem to be the support of autophagy (that naturally tends to reduce with age) at its basic level, including ameliorating metabolic and cellular equilibrium and stability, along with the opposition to excess ROS generation, with the consequent mitigation of inflammatory status [112]. The literature also proposes the establishment of “a direct link between exercise and microbiota gut-brain communication”, with the immune system being an “essential modulator” [113].

There is “moderate to limited evidence” that aerobic exercises (alone or associated with other techniques) in more complex multimodal paradigms are suitable for AD patients [114]. The “multimodal exercise program” type of procedure is frequently considered in the literature, because it mainly attempts to help elderly people adhere to “physical activity and exercising”, with the aim of maintaining or even enhancing muscle strength, endurance, equilibrium, and maybe even ameliorating “stretching” [94]; this would produce favorable effects in terms of the support of physical and cognitive abilities. There is “moderate evidence” for the mitigation of “neuropsychiatric symptoms”, but there are still improvements in both QOL [114] and “functional capacity”, namely, the activities of daily living (ADLs) [115]. There is a possible virtuous circle. The better the self-efficacy and associated confidence, the greater the adherence, which is augmented with every new related level attained in terms of practicing physical exercise [116]—including with the use of technology “through exergames” [94], especially those associated/based on a “game narrative approach” [117]. In fact, “multi-component exercise” [118] or “multidomain interventions”, i.e., encompassing two or more such procedures, ”may have even greater benefits than cognitive training or exercise alone” in elderly individuals. This is especially the case with MCI [106] or other single (including preventive) interventions, this being the case for persons at risk for AD dementia, too [82,119]. However, “the importance of walking outdoors, and PWD (persons with dementia–o. n.) reported physical activity as means of maintaining personhood” [20]. Additionally, on the one hand, physical activity/exercise is, in principle, beneficial no matter the age, even at low intensity, and on the other, therapeutic–rehabilitative procedures involving directed physical exercises can also be considered, including therapeutic sports and music procedures/melotherapy, because these approaches are underpinned by many international trials. Furthermore, combined with rhythmic auditory stimulation (RAS), physical exercise/activity is responsible for beneficial effects on the raw physical force and functionality and smooth coordinated/skilled motions, especially in “mild to moderate AD” [120]. Regarding AD motor impairments, RAS might be increasingly considered, including among the motor neurorehabilitation interventions aiming to improve balance and orthostatic stability, together with walking [121]. Physical exercise, but not music, may produce favorable outcomes in dementia-related “neuropsychiatric symptoms (NPS)” [122]. Additionally, physical exercise associated with education seems to “effectively ameliorate older adults’ depressive symptoms” [123]. But, despite the opinions, in the majority of cases, physical exercise has been shown to favor the clearance of “bioactive substances” (called “exerkines”, e.g., Nerve Growth Factor (NGF), Brain-Derived Neurotrophic Factor (BDNF), Vascular Endothelial Growth Factor (VEGF), Insulin-like Growth Factor 1 (IGF-1), and Adiponectin), which helps to support and ameliorate physiologic and cerebral capabilities [124,125]. Moreover, voluntary physical exercise/training could favorably influence hippocampal-related cognitive function in AD through BDNF (largely considered to be protective) and postsynaptic density protein 95 (PSD-95) modulation in the actions of reactive astrocytes (which, aside from microglia, are the main cellular populations involved in AD pathophysiology [120,121,122,123]. Likewise, AD is “probably identified as age-related impairment of AHN” [119] (adult hippocampal neurogenesis) according to the literature, and it “has been discussed for decades, but there are still inconsistent views on the effect of its intervention in different studies”. More precisely, it is thought that “short-term (2–5 months) physical activity interventions” would be a reasonable formula to favorably address the overall QOL in AD patients based on improvements in cognition and neuropsychiatric impairments. However, regarding their effectiveness, no objective distinction has been found between the different types of interventions [126] or the related outcomes in control groups receiving common care [127]. There are also rather conflicting opinions in the literature regarding the effects of physical exercise on mental activity in AD [13]. Additionally, in regard to this physiatric type of intervention, “the dose-response association” still needs to be clarified, and therefore more research is required in the related methodology domain [128] of this otherwise-considered “possible disease-modifying therapeutic approach” [129] (i.e., “sport is medicine”) [110]. So, we can reasonably conclude that physical activity exercise contributes to AD prevention based on sustaining mitochondrial biological homeostasis and related functioning [130], and overall, physical exercise/activity is an effective “and safe add-on therapeutic intervention” [131], including for patients with AD [131,132]. A particular type of such intervention is the “traditional Chinese medicine (TCM) exercise therapies… (Baduanjin exercise, Tai Chi, Liuzijue exercise and finger exercise)”, which can also “improve MCI” [133], and in general, “traditional Chinese health exercises (TCHEs)” have capabilities “for managing cognitive decline” [134].

#### Music and Dancing

It is considered that listening to one’s preferred music and/or singing and using musical instruments may be able to induce positive feelings, and this effect targets the intimate level of neuroplasticity. In the literature, the actions of listening to music are said to have effects “on autobiographical memory, emotion, and cognitive function in patients” with AD [135]. There is not yet a complete understanding of its beneficial biomedical and psychological effects, but the proposed actions are said to focus on the cerebral structures involved in emotivity and decision functionality, “including sympathetic arousal and dopaminergic circuit activation” [13]. However, generally, melotherapy (neurologic music therapy) is considered in some of the literature data to have the capability of mitigating the expansion of neurodegenerative pathologies such as AD, including combating the related reduction in social relationships and possibly even actions like ameliorating motor impairments that frequently appear after brain lesions [136]. Therefore, the enhanced availability of music is to be considered in domiciliary settings such as “daycare centers and nursing homes” [137]. Music “combines science and art”, and therefore its actions on brain activity can be measured using neurophysiological tests (electroencephalography (EEG), functional near-infrared spectroscopy (fNIRS), and imaging (using functional magnetic resonance imaging (fMRI)). The assessment using these methods shows that there seems to be no important difference between the stimulatory cerebral actions induced by listening to a favorite or an unfamiliar piece of music, while the cerebral activity evaluated with fMRI in both patients with depression and those without “under positive and negative music stimulation” showed “that their regions of interest (ROI) characteristics are quite different” [138]. Regarding the contribution of AI-aided fMRI, including in the approach used for AD patients, it is believed that this could offer useful information for clinicians as additional data for training [139]. According to recent sophisticated neurophysiological and imagistic data, it has been reported that listening to music seems to interfere with cerebral connectivity, thus possibly providing therapeutic benefits to patients with disorders of the default mode networks (DMNs) and/or of the psycho-cognitive reward processes, including in AD. In this respect, “the mPFC (medial prefrontal cortex–o. n.) and PCC (posterior cingulate cortex–o. n.) are most sensitive to changes in functional connectivity” [140]. Listening to music appears to quickly improve “autobiographic memory and category fluency. … some improved in short-term memory, working memory, total verbal recall, and digit span” in people with dementia and in “orientation, and psychomotor speed” after up to four months and six months, respectively, with this kind of procedure [141]. Additionally, listening to instrumentally produced melodies and rhythms, administered for a period of about four months, was shown to consistently reduce hallucinations [9]. There is evidence that patients with AD perform better when words are associated with music [142], and this may even “foster a sense of connection, communication, relaxation, and emotional well-being” [4], aside from the beneficial effect in situations when habitus related to depression and/or anxiety occurs [143] in such patients. However, it should be noted that this is confirmed for severe habitus, whereas the amelioration of memory and/or speech fluency only occurs in mild AD [144]. There is little peer-reviewed published data on the benefits of dancing in the elderly, although there are many reports of improvement in mood, cognition, and coordination. One large prospective study performed in the USA demonstrates that dancing and playing musical instruments are associated with a lower risk of developing dementia. In fact, a so-called active dimension (involving participation vocally/instrumentally and even creating songs and/or dancing) has been identified in contrast to a passive one, i.e., only listening to music [4,13]. Recent literature data suggest the possible prophylactic role of music regarding neurodegeneration in predisposed individuals, considering that a suitable target population is represented by persons either with related genetic risk factors and/or with incipient intellective regress, and the subclinical occurrence of AD usually precedes the full disease [145]. Also, in AD’s incipient stages, to counteract the “impairment in the central executive system” and to improve their “executive functioning”, such older affected persons “may require interventions that are more cognitively intense than traditional” ones. A related example reported in the literature is “a dual-task-based music therapy intervention that involved drum playing and singing”, thus aiming at ameliorating “attentional and motor controls”, with good outcomes [146]. Yet, despite the overall favorable effects of MT, including in AD patients, it has also been observed that “a wide and heterogeneous range of MT techniques” have been reported/approached in various papers; therefore, “this heterogeneity may affect the results of different studies” [147]. Still, a specific mention should be made for the dancing kind of therapeutic intervention: this approach might reduce intellective decline because the motricity driven by music to dance is increased [148]. Moreover, only a few months of practicing aerobic dance may ameliorate episodic recollection in patients with MCI, and this could be organically due to the volume enhancement of the (right and total) hippocampus; this is also because “MCI, especially amnestic MCI (aMCI)”, represents “an intermediate state between normal aging and dementia”, and therefore this is an important target for the preventive dimension of this type of procedure in AD [149].

### 3.5. Information Technology

A general concept sustained by the World Health Organization (WHO) acknowledges that “technology can be used to empower PwAD” (people with Alzheimer’s disease), thus enabling them to have an overall better QOL [150]. Aside from the use of different technological devices like “ the internet and computer”, “texting or videoconferencing among family and friends”, “email”, “telephone”, “social robot”, augmented reality/virtual reality (AR/VR) [87], “tablet applications”, and for a more comprehensive “neuropsychological” evaluation, ”robotic interfaces and wearable sensors”, the additional involvement of “music therapy” could also be more engaging for people with dementia and their caregivers as well, resulting in an improvement in care provision [150]. A large and diverse field of consequent practical applications is grouped as “mobile health (mHealth) technologies”, dating back to 2003 [151]; the Global Observatory for eHealth (GOe) “defined mHealth or mobile health as medical and public health practice supported by mobile devices”. This category includes different wireless supervision/watch devices, personal digital assistants (PDAs), and even smart mobile phones [152] that may serve as therapeutic interventions (e.g., art-based interventions, reminiscence therapy, cognitive training therapy, and mentalizing imagery therapy) [151]. Types of interventions such as “computer-based (or ”computerized”–o. n.) cognitive training (CCT)” [153] appear to be “a potential instrument for the improvement of cognition”. In fact, it is a technical facility that enables people to use their digital devices, including mobiles, to become interactively involved in intellective practice, possibly with access to some elements of VR (which provides especially good results), with feasible and measurable outcomes including noninvasiveness and accessibility (no special training needed and rather inexpensive). These approaches work well with technical “standard criteria” and “sustainability”, especially regarding global cognition and, specifically, “working memory, executive function … processing speed” [154]. Also, the use of the “digital-app version of the Photo-Activity”, especially of the “person-centered artistic” kind, might be a sort of “psychosocial intervention”, helpful for both cognitive-behavioral and social relationships among institutionalized patients with dementia, including those with AD [155]. The same seems to go for “digital storytelling”, which represents a rather newer computer-based facility, and could improve QOL through tracing and reciprocating various lived situations, including for elderly people with AD [156], “after having viewed their story” [157]. This may partially be true for telemedicine as well, especially in situations that lack opportunities for a direct professional approach [158], as the majority of the procedures based on mHealth technologies seem to provide good results when availed by patients with MCI or dementias, including AD [159]. Digital/information technology also raises some issues related to human-computer interaction (HCI). Hence, on the one hand, in order to maximize its bio-medical and social advantages and, on the other hand, to minimize possible side effects (for instance, different degrees of dependency/addiction), an adequate approach to this rapidly developing domain focuses on the “positive technology” paradigm. This focuses on facilities meant to augment the beneficiary’s experience and mainly encompasses ”hedonic technologies”, used to generate good/beneficial feelings, ”eudemonic technologies”, which help people to become involved in and achieve life experiences, and “social/interpersonal technologies” that sustain fair inter-human (individually and/or collectively) relationships. All of these are based on providing different digital solutions, including “virtual reality environments”, targeting psychological soundness and QOL [160]. Yet, a precise rating of these already numerous and rather diverse kinds of digitally based technologies is difficult, at least regarding one of their main scopes: to mitigate loneliness and/or social isolation, with their negative aforementioned consequences, considering “the vagueness of the concepts and related measures” [87].

#### Virtual Reality (VR)

The developing domain of VR/VE (virtual environment) interventions also encompasses augmented reality (AR) procedures. As advanced digital facilities, regarding their therapeutic–rehabilitative outcomes, all of them depend on both the technological types and level of the devices used and the potential beneficiaries’ capabilities to favorably interact with the kinds of information such methods provide [161]. Although rather old, appearing in the 1980s, VR boosted its involvement in addressing different pathologies, including neuropsychiatric types like AD, at the beginning of the last decade, partially because of related concerns regarding the neuro-/biopsychological side effects [162,163]; specifically, some patients with AD “did experience boredom, fear, and anxiety while using VR applications” [164], as well as possible social, more extended, consequences. Therefore, generally speaking, from a medical perspective, since the beginning VR has been considered as a bundle of modern methods that can be procedures added on to the classical therapeutic–rehabilitative ones, without tending to replace them, which can (if appropriately indicated and applied) supplement the multimodal information provided to the users; it thus extends the amount of data brought to them, sometimes more safely and/or ecologically, and hence, it may offer a better-matched construct of the specific sanogenic approaches’ administration [165]. In particular, it can help through its immersive dimension/component, an advanced type of apparatus consisting of head-mounted displays (HMDs) that are able to virtually reproduce/simulate real-life environments and situations, which interact with the perceptions/related experiences of “the first-person perspective (1 PP)”, thus being able “to facilitate recovery and enhance motor or cognitive functions in” such affected persons, with applications that include reproducing real-life situations (e.g., shopping with a pre-established list of products and visiting different virtual market environments); this is a training and measurable approach, interacting with important challenging cognitive functions, such as working recollection, spatial memory and orientation, intellective planning, and related performance [166], with favorable effects on cognition preservation [167]. It should also be noted that the mitigation of apathy has been observed more after “non-immersive virtual experiences” [168]. However, there are also more reticent opinions regarding the safety and nuanced effectiveness of the use of VR in AD patients [168,169]. On the other hand, targeting a very desirable therapeutic direction to be attained, i.e., relaxation and positive emotions provision, in the literature, the most common real-world elements reproduced for this purpose, by (immersive) VR, are short movies displaying beautiful outer and/or inner natural landscapes [170]. From intimate and motor points of view, in AD this kind of intervention would both induce neuroplasticity and support pace practice by matching plural digital make-believes of routine activities with an exercising gait on a treadmill and/or on a stabilometric-type moveable apparatus, meant to (re-)train stance and motion [171]. To the above examples of beneficial VR interventions must be added another, considering that elderly patients often have more or less severe disabilities [163,170] (not seldom cumulative, as the characteristic pathology paradigm in the elderly is multimorbidity [172]) that limit or can even jeopardize their access in natural environments [163,170]. Complementary to what we have pointed out, there are no significant differences reported in perceiving concrete vs. digitally achieved backgrounds, regarding physical parameters like eye blinks and heartbeats, heart rates, and electrophysiological rates, such as frequency band-powers, collected electroencephalographically [170]. Newer literature data suggest a “next-virtual” level would attempt to merge the “simulation technologies”, i.e., “virtual reality (VR), augmented reality (AR)” and a “mixed” one, towards achieving a so-called “extended reality (XR)”, whose “core” would be “metaverse”, based on “haptic technologies embedded”, using detectors or actuators, in order to reproduce haptic perceptions autonomically (e.g., body temperature) and/or somatically (forces interaction), etc., that the concrete milieus generate in individuals [173]. Actually, in AD, the rousing effect of rehabilitation interventions, with beneficial actions towards recovery after cerebral lesions, seem to “have shown promising results with XR-assisted treatments” [174]. At the same time, VR enhances “the fun and enjoyment of daily physical activity” (physical activity is for the elderly a basic natural, and in principle very accessible, endeavor, including for sickness and consequent disablement prophylaxis [175], boosting the patients’ related engagement, especially associated/based on a “game narrative approach” [117], within the already well-known concept and a bundle of such related type of interventions, so-called “serious games” [176].

### 3.6. Lifestyle Factors

Factors such as diet and exercise improve cognitive function and may protect against AD [177]. More precisely, there is a quite broad spectrum of opinion regarding the beneficial effects of some of the Mediterranean diet’s components and in the complete diet itself in counteracting intimate processes of aging (in general and of the brain in particular) and in mitigating the related intellectual decline, including AD. The same seems to be true, on one hand, for the ketogenic [178] diet, “a dietary approach characterized by high-fat and low-carbohydrate intake, aiming to facilitate weight loss, enhance mental clarity, and boost energy levels” [179], and for physical activity/exercise, on the other [178]. Yet, it is still not clear whether a change in diet in middle-aged or elderly individuals can improve or halt the progress of mild cognitive MCI to dementia [180].

#### 3.6.1. Sensory Practices

Within this generic denomination are included “aromatherapy, massage, multi-sensory stimulation, bright light therapy)” [20]. The literature suggests that these provide good outcomes for some often-encountered problems in AD, e.g., habitus problems and symptoms of depression [143].

#### 3.6.2. Validation Therapy

This kind of nonpharmacological intervention is part of what some authors collectively denote “psychosocial practices” (aside from, for instance, “meaningful activities”, “pet therapy”, as well as reminiscence therapy (RT) and “music therapy” [20]; however, we consider the latter two to be more consistent; therefore, we placed them into a different taxonomy–see “Music and dancing”). However, such procedures can be also ”an alternative approach to treating delusions and hallucinations in dementia” [9].

### 3.7. Low-Dose Ionizing Radiation (LDIR)

Lately, in the literature, low-dose ionizing radiation (LDIR) has been considered to be a kind of procedure adequate for use in AD, among other neurodegenerative diseases. This is connected to its possible intimate actions: “radiation hormesis”, which refers to a general capability of living entities to favorably respond by adaptation to various stressor agents and, hence, to enhance their biological resilience and functional performances, this being the case of LDIR, too [181]. More precisely, it supports enzymatic repair in various biological structures, including the deoxyribonucleic acid (DNA) level, with favorable actions on the gene expression related to neuroprotective biological mechanisms, within related protective mechanisms of antioxidant prophylaxis, with ROS depletion and enhancement of the cells’ resilience to the damaging actions of these moieties, and the consequent convergent promotion of the genome’s stability, with anti-inflammatory (inclusive regarding neuroinflammation) actions in various tissues. As for the central nervous system (CNS), at least mainly in animal experimental models, it would promote molecular and synaptic entities’ performance and even myelin production and neurogenesis. Besides these many beneficial effects, an open question remains regarding the possible appearance of detrimental actions of applying LDIR to the CNS [181].

### 3.8. Mechanical-Based Stimulation

As found in the literature, this kind of different procedural intervention mainly encompasses whole-body vibration (WBV), transcranial ultrasound stimulation (TUSS), and lately, auditory stimulation (AS) [182]. Regarding WBV, this represents a kinesiologic–electromechanically-based type of therapeutic–rehabilitative and/or even prophylactic procedure that comprises the exposure to low-frequency mechanical vibrations, using a related apparatus able to produce such a form of energy, usually a kind of specific platform on which the whole body is placed [183]; it is a mechanical low-frequency vibratory passive stimulation [184] that “may offer an alternative for active exercise training... “ proving to be “an effective intervention” with beneficial actions on various body structures and functions “to improve physical fitness” mainly of “the musculoskeletal system...” but also with favorable involvement at the hormonal and nervous system levels, specifically through favoring neurotrophy, neurotransmission, and maybe even neurogenic functions, all resembling the effects of active exercises [185]. In animal experiments, TUSS has shown favorable actions on cerebral circuitry and afferent neuroplasticity; specifically, AS, based on using “trains of tones”, appeared to sustain gamma brain waves, with beneficial effects in AD at the intimate pathological level and consequent related habitus improvement [145]. Generally, the three aforementioned types of “low-cost and noninvasive mechanical-based interventions” have shown their “effectiveness, safety, and feasibility” in the complex symptomatic spectrum of AD, but because the related human studies are still few and heterogenic in terms of the therapeutic methodology used, there is need for further confirmation [182].

### 3.9. Photobiomodulation (PBM)

This physiatric type of intervention, also called (low-)level LASER therapy (LLLT) and LASER biostimulation, also seems to be promising in AD. Its main beneficial effects are based on a claim in the literature that it improves action at a mitochondrial level [186]; it ameliorates the “behavioral results and reduces amyloid plaques and neurofibrillary tangles”, seeming especially to have favorable actions at the basic intimate mitochondrial level including “mitochondria fission and fusion” regulation (of “glial cells and neuroinflammation”, too), stimulating/modulating the cytochrome c oxidase (CCO) functionality, and through antioxidant actions. On the other hand, different collateral or long-term (side)effects need further examination, as the “brain is a difficult-to-irradiate organ” [187].

### 3.10. Reminiscence Therapy (RT)

This intervention, defined as “structured use of memories, experiences, and prompts” [188], was considered for the cognitive treatment of AD, beginning in the 1990s, and it specifically consists of stimulating the affected persons to, as much as possible, accurately retrace and interactively communicate memories of their lived experiences [189], thus being an “appropriate nursing intervention to the cognitively impaired elderly” [190]. The construct particularly of the RT is the preponderant emphasis on remote memory, with less importance given to the short term, by both the AD patient and the “facilitator” [190]. Generally, RT is a nonpharmacological care type of intervention that may be used in order to overall ameliorate the QOL and as an additional procedure to mitigate depression and anxiety [189]. Basically, RT is a category of “cognitive therapy” interventions, usually associating/incorporating “reality orientation training” [30]. Works on this subject have shown the beneficial value of RT, aside from the abovementioned fields, on intellectual functions, including an improved capability to fulfill tasks of daily living, based on recalling their former memories, mainly in mild-to-moderate AD patients, in association with relevant related static and/or moving images and music [191], as a choice/complementary intervention for “treating delusions and hallucinations in dementia” [9]. In addition, regarding RT, “more recently, digital storage and presentation of photographs, music, and video clips have become widely used [192]. An important biomedical link has been reported between RT and feeding; the former encourages persons with dementia to enjoy meals specific to the origin culture and remote experiences from their childhood, including participating in traditional related gastronomic events and involvement in cooking, using their tastes and customs, established before becoming mentally ill [10]. Moreover, through ”electroencephalography (EEG) signals for automatic emotion recognition” one can see the objective effects of RT; in addition, “the pleasure level of RT and the (o. n.: eu-)stress level of the conversation is more conducive to the emotion classification of older people in the communication support systems” [193]. Hence, RT “can help improve communication, feelings of belonging..., mood, wellbeing”, with beneficial intellective actions; further, with the aid of the impressive actual development and expansion of artificial intelligence (AI), there can be related digital progress: re-configuring RT in a customized paradigm, resembling the features provided by a human health professional (skilled/licensed carer or respectively, therapist), targeting thereby an “intangible cultural heritage”, including “facial expression analysis and reinforcement learning techniques”. Thus, by recognizing a user’s facial emotions, an adequately informed piece of software could emulate RT more appropriately and promptly [194]. AI can thus be helpful “for assessment of cognitive and functional impairment in” AD patients [195]. Apart from cognition, RT could thus improve the QOL of patients with dementia, including AD and/or vascular [196], and “using immersive VR may be more effective as it would be more realistic than traditional reminiscence therapy and could lead to increased engagement”, proving, according to different works, to be effective against “anxiety, depression, apathy, and negative mood states” [170], as well as “improving semantic verbal fluency, immediately after a short intervention program in elderly”^,^ and even reducing “depressive symptoms” [197]. In this context, we emphasize the important relation between the appearance of cognitive involvement/impairment and depression; in the elderly, the rate of intellective downfall can actually also predict the appearance or ingravescence of (associated) depression: [123]. Conversely, regarding depression (and anxiety, too), it “may be clinically observed many years before the onset of significant cognitive symptoms” [82].

### 3.11. Repetitive Transcranial Magnetic Stimulation (rTMS)

This is a physiatric procedure/intervention that appears promising in AD, with favorable pathogenic targeting outcomes, i.e., diminishing the accretion of Aβ peptides, counteracting “tauopathy”/”hyperphosphorylation”, and, at the geno-molecular level, mitigating ApoE “expression” and stimulating (protective) autophagy [187]. An “open-label extended follow-up study” claimed that multisession maintenance, with the therapeutic administration of rTMS, sustained over the long term, was preferable to a short-term dosage (for instance, two weeks in AD). In addition, it may be better when “multisite” is administered, mainly acting on “cognitive and executive functions” [198]. Although not well understood [199,200], its principal known action mechanisms are as follows. In AD, the amount of BDNF dependent on/controlled by the long-term potentiation (LTP) and the overall functioning of neurons decreases in the rTMS, thus increasing the levels of this neurotrophic factor (since a longer time is known to have neuroprotective effects in AD [179], as well as in PD [180,201]). Redressing or at least abating abnormal LTP-like neuroplasticity and the connected disturbances of cells’ communication bio-codes and “concurrent cognitive training and/or patients with higher education” may result in better outcomes” [200]. An important genetic factor involved in the predisposition to sporadic AD is the presence of the APOE ε4 allele, which would detrimentally interfere with the gamma-aminobutyric acid (GABA)’s -ergic deterrent support connectivity, thus altering the aggregation of the Aβ peptide and also the egestion of its soluble form. Accordingly, “rTMS as a modifier of inhibitory neuron function… reduces GABAergic synaptic strength on principal neurons” [199]. So, as rTMS is “an inhibitory neuron function modifier” and including its action on GABAergic synapses, it can favorably intervene in the functional ensemble of the equilibrium between neural inhibition and excitation [200]. Aside from modulating synapses’ activity/neuroplasticity and exerting antiapoptotic actions, as well as PBM, it regulates neuroinflammation and, although not fully confirmed, on glial cells. On the other hand, rTMS needs further research, including a more complete identification of possible undesirable reactions, among which are (although rare) seizures [187].

### 3.12. Transcranial Direct Current Stimulation (tDCS)

Aside from the dose, this kind of physiatric intervention has polarity dependence, stimulatory under the anode and inhibitory under the cathode, in terms of its effects (there remains debate regarding their intimate electro-physiological support and consequent real actions) [187,202,203]. Through ample and complex actions, tDCS (mainly under the positive electrodes) exerts favorable effects on cognition/learning, visual recognition, spatial, word, and working types of memory in AD (Figure 2). At intimate levels, it combats amyloid/Aβ peptide plaques depositing (as opposed to their consequent vicious circle with the astrocytes’ activation), ameliorates cerebral circulation, improves synapses’ plasticity, and stimulates N-methyl-D-aspartate (NMDA) receptors, also having anti-neuroinflammation actions [204].
ijms-24-16533-t002_Table 2Table 2Nonpharmacological interventions in AD: a synthetic description of cellular/tissue and molecular aspects.*Nonpharmacological Interventions in AD**Cellular/Tissue and Molecular Aspects**References**Acupuncture**Acupuncture in AD potentially modulates neurotransmitters and neuroinflammation, improving ADAS-cog and CIBIC-Plus scores. Data mining aids acupoint selection.*[78,79,80]*Electroacupuncture**Electroacupuncture enhances MoCA scores in AD, possibly via cellular mechanisms such as neuroplasticity and molecular pathways like neurotransmitter modulation.*[78]*Cognitive behavioral therapy (CBT)**CBT-I may improve cognitive function and potentially delay AD onset through the modulation of Aβ deposition at a molecular level.*[81]*Counseling and psychoeducation**The intervention promotes healthy lifestyles and better coping strategies, with a potential cellular impact on BDNF expression, affecting mental health in mild AD.*[82,83,84,85]*Environment adjustment**Interventions to combat loneliness may impact regional WM density in the brain cortex, potentially affecting AD development. Activities like aromatherapy with Salvia officinalis could influence acetylcholinesterase levels, offering molecular-level therapeutic potential.*[86,87,88,89,90,91,92,93,94,95,96,97]*Exercise/Physical activity**Exercise-based nonpharmacological interventions show multifaceted benefits across cellular and molecular domains, impacting homeostasis, transcription, protein function, and BDNF release. Physical exercise modulates brain metabolic activity, enhances the release of Brain-Derived Neurotrophic Factor (BDNF), and possibly influences monocyte functions, potentially offering a holistic approach to managing AD. Exercise also affects cerebral blood flow and mitochondrial function, although its precise efficacy remains under debate.*[13,20,82,94,96,97,98,99,100,101,102,103,104,105,106,107,108,109,110,111,112,113,114,115,116,117,118,119,120,121,122,123,124,125,126,127,128,129,130,131,132,133,134]*Music and dancing**Music-based interventions, including active participation and passive listening, impact neuroplasticity, evoke positive emotions and influence cognitive function in Alzheimer’s disease (AD) patients. These interventions modulate brain structures linked to emotivity and decision-making via dopaminergic circuits and sympathetic arousal. Technological aids like functional MRI and EEG indicate differential neural responses to music, suggesting utility in AD management. Molecular changes, although not fully understood, appear to involve neurotransmitter pathways and neuroconnectivity, particularly in regions like the medial prefrontal cortex and posterior cingulate cortex.*[4,9,13,135,136,137,138,139,140,141,142,143,144,145,146,147,148,149]*Information technology**Technology, including mHealth and computer-based cognitive training, empowers Alzheimer’s patients and caregivers by enhancing cognition and psychosocial well-being. These interventions likely modulate neural pathways and cellular functions, although the exact molecular mechanisms remain underexplored.*[87,150,151,152,153,154,155,156,157,158,159,160]*Virtual reality (VR)**VR and AR interventions in Alzheimer’s Disease (AD) likely impact neuroplasticity, cognitive function, and motor skills by modulating neural pathways. These immersive technologies may interact with molecular markers associated with cognitive and emotional regulation, although specific mechanisms warrant further study.*[117,161,162,163,164,165,166,167,168,169,170,171,172,173,174,175,176]*Lifestyle factors**Dietary interventions like the Mediterranean and ketogenic diets, as well as exercise, are suggested to modulate cognitive function possibly through anti-inflammatory and antioxidant pathways, impacting cellular and molecular mechanisms relevant to Alzheimer’s Disease. Further studies are needed for mechanistic insights.*[177,178,179,180]*Sensory practices**Alternative therapies like aromatherapy and light therapy may modulate neurotransmitter levels and circadian rhythms, potentially ameliorating behavioral symptoms in Alzheimer’s Disease.*[20,143]*Validation therapy**Psychosocial interventions may influence neurotransmitter systems and neuronal plasticity, potentially alleviating delusions and hallucinations in dementia patients.*[9,20]*Low-dose ionizing radiation (LDIR)**Low-dose ionizing radiation (LDIR) in Alzheimer’s treatment may induce radiation hormesis, enhancing DNA repair, gene expression, antioxidant defense, and anti-inflammatory actions, while potentially improving synaptic and myelin integrity.*[181]*Mechanical-based stimulation.**Whole-body vibration (WBV), transcranial ultrasound stimulation (TUSS), and auditory stimulation (AS) may impact neurotrophic and neurotransmission pathways. WBV enhances musculoskeletal and hormonal systems, TUSS affects cerebral circuitry and neuroplasticity, while AS modulates gamma brain waves.*[145,182,183,184,185]*Photobiomodulation (PBM)**Low-level LASER therapy (LLLT) in Alzheimer’s Disease (AD) mainly targets mitochondrial function, modulating cytochrome c oxidase (CCO) and exerting antioxidant effects. It also impacts mitochondrial fission/fusion and neuroinflammation.*[186,187]*Reminiscence therapy (RT)**Reminiscence therapy (RT) in AD focuses on stimulating remote memory. While not directly cellular or molecular, EEG signals suggest its neurophysiological relevance, impacting cognition and mood.*[9,10,30,82,123,170,188,189,190,191,192,193,194,195,196,197]*Repetitive transcranial magnetic stimulation (rTMS)**Repetitive transcranial magnetic stimulation (rTMS) in AD shows promise at the cellular and molecular levels by reducing Aβ peptides, counteracting tau hyperphosphorylation, and modulating ApoE expression. It also influences BDNF levels and GABAergic synaptic strength, potentially rectifying imbalanced neural inhibition–excitation dynamics.*[179,180,187,198,199,200]*Transcranial direct current stimulation (tDCS)**Transcranial direct current stimulation (tDCS) in Alzheimer’s Disease (AD) shows polarity-dependent effects, enhancing cognition and combating Aβ peptide deposits. It improves cerebral circulation, synaptic plasticity, and NMDA receptor activity, while also exerting antineuroinflammatory effects.*[187,202,203,204]

## 4. Discussion

Overall, we underline the heterogeneity, responsiveness, and adherence, especially of the “oldest olds” [205], to these kinds of interventions, very much depending on the patients’ general biological reserve and their effective neuroplasticity.

### Limitations

A large amount of related literature—116 selected articles issued in (only) one year (2022)—is, on one hand, encouraging as regards the legitimate growing interest in this subject matter, but this comes along with the heterogeneity of the literature data; hence, we cannot guarantee the lack of bias within these data. As we emphasized, we taxonomically grouped the nonpharmacological interventions used in the treatment-rehabilitation of the AD patients either “classically”, i.e., with a relatively logical paradigm, considering either the same or neighbor category of energy used or with a common focus on a different target bundle of AD symptoms. However, we put these together asserting that as nonpharmacological interventions, particularly in AD, are quite diverse and at least some of them are either mixed and/or rather nondescript, we have also arranged them in tabular form by their names, that is in alphabetic order.

## 5. Conclusions

As AD has a growing frequency, with the increasing global demographic aging process, and still there is no cure, we considered different approaches aiming to improve the current therapeutic–rehabilitative outcomes, including nonpharmacological endeavors (possibly strengthened by artificial intelligence, as this is largely promising but also, as wide discussed in the topical literature, has risks) are justified, and their periodic reappraisals welcome. The same is true for both fundamental research and clinical trials, in order to improve and hasten the awaited translation from bench to bedside.

Accordingly, the aim of our systematic literature review (i.e., a reappraisal of the actual data regarding AD exhaustively, from the basic mechanisms behind its development to the main current clinical–epidemiological features and to the newer nonpharmacological therapeutic–rehabilitative interventions considered in the literature, with their principal cellular/tissue and molecular actions) is to provide topically related information, hopefully, useful in such an interesting and evolving domain.

## Figures and Tables

**Figure 1 ijms-24-16533-f001:**
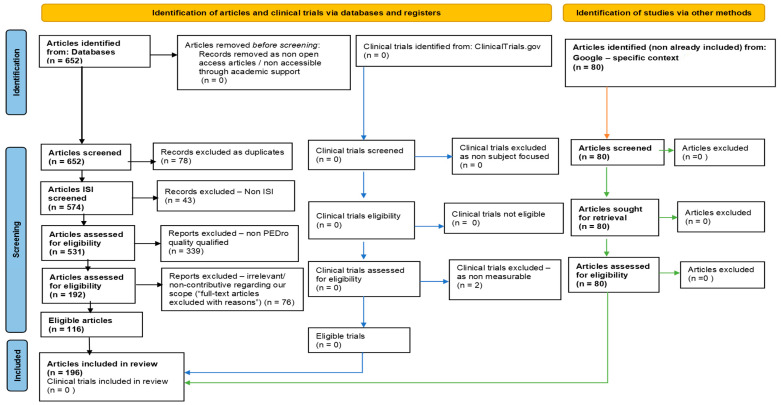
PRISMA-type flow diagram of our systematic literature review.

**Figure 2 ijms-24-16533-f002:**
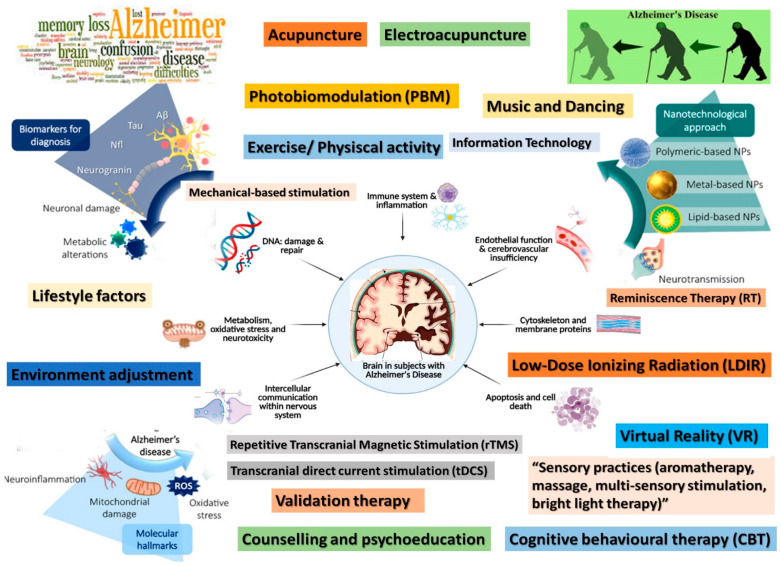
Main nonpharmacological interventions in AD, identified in the literature, are used within this systematic review, as some possible insights on their principal cellular and molecular actions.

**Table 1 ijms-24-16533-t001:** Keyword combinations/syntaxes used contextually within our systematic literature review search.

Keywords	Elsevier	PubMed	PMC	PEDro	Total
“Alzheimer’s disease” + “Video game therapy”	0	0	0	0	0
“Alzheimer’s disease” + “Augmented reality therapy”	0	0	0	0	0
“Alzheimer’s disease” + “Virtual reality therapy”	0	0	19	0	19
“Alzheimer’s disease” + “Serious games therapy”	0	0	0	0	0
“Alzheimer’s disease” + “Reminiscence therapy”	1	6	96	0	103
“Alzheimer’s disease” + “Music therapy”	0	16	249	0	265
“Alzheimer’s disease” + “Dancing therapy”	0	0	0	0	0
“Alzheimer’s disease” + “Exercise therapy”	2	22	241	0	265
Total	3	44	605	0	652

## Data Availability

Not applicable.

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
