# Peer review of "Topical Cellular/Tissue and Molecular Aspects Regarding Nonpharmacological Interventions in Alzheimer’s Disease—A Systematic Review"

_ijms, 2023, doi:10.3390/ijms242216533_

Round 1
Reviewer 1 Report
Comments and Suggestions for Authors
In this review manuscript, Aurelian et al., highlights the importance of ongoing efforts to improve therapeutic, rehabilitative, and preventive approaches to address the challenges posed by Alzheimer's Disease, especially since there is no cure for this disease. The topic is relevant, and work highlights the urgency of addressing the growing impact of dementia, especially AD, as the population ages. There are several concerns that make this manuscript unsuitable for publication in a reputed journal like IJMS.
1. The English language should be improved, and the manuscript needs a thorough grammar check. The sentences are extremely long and confusing. I suggest that the authors to contact a professional editing service.
2. Throughout the text, the authors have used quotations that affects the readability of the sentences and disrupts the flow of the manuscript.
3. The abstract is written in complicated language and needs to be edited. For example, the sentence “This reality has as one of the harsh medical, social, and economic consequences of the continuously increasing number of people with dementia, including Alzheimer's Disease (AD), accounting for up to 80% of all such types of pathology” is confusing.
4. The introduction needs to be restructured. The information about existing knowledge gap, how this review addresses the gap, and the novelty of this manuscript is missing from the introduction.
5. Rephrase the sentence “Consequently, the number of people living with dementia, which comprises ”more than 55 million people”1 – Alzheimer's disease (AD – after the name of Dr. Alois Alzheimer) representing up to 80% of all dementias – augments (for instance: over 6 million persons are affected in the USA (3,4); it is ”the most/ main common form/ type/ cause of dementia” (3,5–8) – and the ”most widely studied” one (9) – (followed by ”vascular dementia, frontotemporal dementia and dementia with Lewy bodies” (10), and if added the ”prodromal AD and ”preclinical AD”, it results ”416 million across the AD continuum” (11)).
6. Some references are missing. For example, “Major risk factors for sporadic AD are increasing age and inheritance of the e4 allele of the apolipoprotein E gene.”
7. Throughout the text, there is no cohesiveness between the paragraphs and the sentences are vague. Results are stated, however their implications in the context of this manuscript is not properly explained.
8. The sections Ab and Amyloid Precursor Protein (APP) should be combined. The authors should expand the section on Familial and Sporadic AD and include the several other existing hypotheses for sporadic AD (Mitochondrial hypothesis for Sporadic AD etc.)
9. What is IPAD? How is this relevant?
10. The sections Acupuncture and Electroacupuncture should be combined. The sections Exercise/Physical activity and Music and dancing should be combined. Lifestyle factors, Sensory Practices and Validation therapy should be combined.
11. Figure 2 should be formatted. It will be informative to summarize the key findings and their implications in a tabular format.
12. In the conclusion section, the authors need to discuss the big picture of the study and the future implications.
Comments on the Quality of English LanguageThere are a lot of typos. The sentences are long and confusing.
Author Response
Dear Peer-Reviewer,
Thank you for your thorough comments and contributive and suggestions for the improvement of our manuscript. Accordingly, please find herein, below, our point-by point answers:
- You are right: Accordingly, we have reviewed both: the English grammar used and we have shortened, the sentences/ phrases and also, made them clearer; so, now, we reckon the manuscript, including from this point of view, has improved. However, if still necessary – as we are not native English speakers – we shall contact the IJMS professional editing service.
- Hopefully resolved, i. e. we have done consistent editing, thus reducing the number of quotations. Yet, the use of many such quotations has been – and we still consider appropriate to maintain a part of them – because, as you could determine, not few works present different such non-pharmacological interventions quite enthusiastic, given the fact that practically at present there is no cure for AD, and not even decisively effective therapeutic-rehabilitative interventions able to provide spectacularly improvements (including) regarding this disease; so, using quotations we considered to thus keep a relative distance towards some too enthusiastic assertions.
- Resolved. Please see the revised – according to the Peer-Reviewers’ comments and suggestions – form of our manuscript: all the new added words and/or rephrased sentences/ paragraphs are colored in red, just in order to be easy to be observed.
- Hopefully resolved. Please see the revised – according to the Peer-Reviewers’ comments and suggestions – form of our manuscript: all the new added words and/or rephrased sentences/ paragraphs are colored in red, just in order to be easy to be observed.
- Resolved. Please see the revised – according to the Peer-Reviewers’ comments and suggestions – form of our manuscript: all the new added words and/or rephrased sentences/ paragraphs are colored in red, just in order to be easy to be observed.
- Resolved. Please see the revised – according to the Peer-Reviewers’ comments and suggestions – form of our manuscript: all the new added words and/or rephrased sentences/ paragraphs are colored in red, just in order to be easy to be observed.
- Hopefully resolved. Please see the revised – according to the Peer-Reviewers’ comments and suggestions – form of our manuscript: all the new added words and/or rephrased sentences/ paragraphs are colored in red, just in order to be easy to be observed.
- Resolved. Please see the revised – according to the Peer-Reviewers’ comments and suggestions – form of our manuscript: all the new added words and/or rephrased sentences/ paragraphs are colored in red, just in order to be easy to be observed.
- Please see, first, within the Introduction section (”Soluble waste substances, including Aβ drain from the brain along thin membranes in the walls of capillaries and arteries as Intramural Periarterial Drainage (IPAD)”, and respectively: ”The pattern of distribution of Aβ in the blood vessel walls in both human and mouse CAA suggests that Aβ is eliminated from the brain along Intramural Periarterial Drainage (IPAD) pathways that fail with advancing age and possession of APOE4 genotype” – within the Pathology of AD section.
- Resolved. But, like we have briefly announced since the Results section (”we have marked out for arranging them in two complementary ways: rather ”classical”, i.e., grouped mainly as presented above, and respectively, in tabular form (including) by their names, that is in the alphabetic order of their most frequently found in the literature denominations” and expressed in the Discussion section: ”we have taxonomically grouped the non-pharmacological interventions topically used in the treatment-rehabilitation of the AD patients, somehow ”classically”, i. e. on a relatively logical paradigm – considering either the same or neighbor category of energy used, or a common focus on different target bundle of AD symptoms – but we anneal asserting that as non-pharmacological interventions, including in AD, are quite diverse and – at least some of them – either mixed and/or rather nondescript, we have marked out for arranging them also in tabular form (including) by their names, that is in the alphabetic order of their most frequently found in the literature denominations”. Please see the revised – according to the Peer-Reviewers’ comments and suggestions – form of our manuscript: all the new added words and/or rephrased sentences/ paragraphs are colored in red, just in order to be easy to be observed.
- Resolved. Please see it in the revised – according to the Peer-Reviewers’ comments and suggestions – form of our manuscript.
- Hopefully resolved. Please see the revised – according to the Peer-Reviewers’ comments and suggestions – form of our manuscript: all the new added words and/or rephrased sentences/ paragraphs are colored in red, just in order to be easy to be observed; specifically (in the Introduction section): ”the aim of our systematic literature review is to highlight the basic mechanisms behind the development of AD, in connection with the role of its risk and diagnosis factors, and to use this information for a comprehensive topical reappraisal attempt on the – including newer – considered in the literature, non-pharmacologic therapeutic-rehabilitative interventions, with their intimate, cellular and molecular actions, thus approaching a relative gap of structured knowledge in this domain” and respectively, in the Conclusion section: ”Accordingly, the aim of our systematic literature review, i. e. a reappraisal of the actual data regarding AD, exhaustively: from the basic mechanisms behind its development to the main current clinical-epidemiological features, and – including newer, considered in the literature – to the non-pharmacological therapeutic-rehabilitative interventions – with also, their principal cellular/ tissue and molecular actions – is to provide a topical related information, hopefully useful in such an interesting and evolving domain.
Reviewer 2 Report
Comments and Suggestions for Authors
In this systematic review, the authors summarized the non-pharmacological interventions in Alzheimer's disease published during the year 2022. Overall, the manuscript is quite informative. Meanwhile, there are a few concerns that might significant weaken the manuscript.
1. When the authors cite the findings or statements from previous papers, the authors frequently cite the original sentences by using double quotes, which is inappropriate. Please rephrase and summarize the cited portion to make it more readable.
2. The section "Results and Discussion" should be separated.
3. In the Results part, the authors list the non-pharmacological interventions alphabetically, which is a quite lazy way. The authors might need to group these non-pharmacological interventions in a logical way that makes sense. For example, acupuncture and electroacupuncture should be in one category, information technology, virtual reality (VR) could be in one category, etc.
4. In the title, the authors highlight "cellular and molecular aspects", meanwhile, not all studies with non-pharmacological interventions produced cellular and molecular results. When assessing the effect of the non-pharmacological interventions, did the authors apply any consensus criteria? What are the criteria?
5. The authors might provide a table briefly summarizing the effect of the non-pharmacological interventions in Alzheimer’s disease.
Comments on the Quality of English LanguageThe quality of English language is OK.
Author Response
Dear Peer-Reviewer,
Thank you for your thorough comments and contributive and sugestions for the improvement of our manuscript. Accordingly, please find herein, below, our point-by point answers:
- You are right. Hopefully resolved, i. e. we have done consistent editing, thus reducing the number of quotations. Yet, the use of many such quotations has been preserved – and we still consider appropriate to maintain a part of them – because, as you could determine, not few works present different such non-pharmacological interventions quite enthusiastic, given the fact that practically at present there is no cure for AD, and not even decisively effective therapeutic-rehabilitative interventions able to provide spectacularly improvements (including) regarding this disease; so, using quotations we considered to thus keep a relative distance towards some too enthusiastic assertions.
- Resolved.
- Resolved. Please see the revised – according to the Peer-Reviewers’ comments and suggestions – form of our manuscript: all the new added words and/or rephrased sentences/ paragraphs are colored in red, just in order to be easy to be observed.
- Hopefully resolved: we have added to the title ”/tissue”. But actually, to all the non-pharmacological interventions presented there are discussed the intimate (at cellular/ tissue and molecular, levels) mechanisms of action and consequent therapeutic-rehabilitative effects. ”The heterogeneity of this diverse field matches with the consensus criteria construct, of Delphi ”technique (DT)” (Arakawa N, Bader LR. Consensus development methods: Considerations for national and global frameworks and policy development. Res Social Adm Pharm. 2022 Jan;18(1):2222-2229. doi: 10.1016/j.sapharm.2021.06.024. Epub 2021 Jun 30. PMID: 34247949.), this being the main reason for the muti-professional panel of the authors (physicians: Gerontology and Geriatrics, Physical and Rrehabilitation Medicine, Psychia-try, Neurology, Neuropathology, Neurobiology, IT”. Please see the revised – according to the Peer-Reviewers’ comments and suggestions – form of our manuscript: all the new added words and/or rephrased sentences/ paragraphs are colored in red, just in order to be easy to be observed.
- Hopefully resolved. But, like we have briefly announced since the Results section (”we have marked out for arranging them in two complementary ways: rather ”classical”, i.e., grouped mainly as presented above, and respectively, in tabular form (including) by their names, that is in the alphabetic order of their most frequently found in the literature denominations” and expressed in the Discussion section: ”we have taxonomically grouped the non-pharmacological interventions topically used in the treatment-rehabilitation of the AD patients, somehow ”classically”, i. e. on a relatively logical paradigm – considering either the same or neighbor category of energy used, or a common focus on different target bundle of AD symptoms – but we anneal asserting that as non-pharmacological interventions, including in AD, are quite diverse and – at least some of them – either mixed and/or rather nondescript, we have marked out for arranging them also in tabular form (including) by their names, that is in the alphabetic order of their most frequently found in the literature denominations”.Please see the revised – according to the Peer-Reviewers’ comments and suggestions – form of our manuscript: all the new added words and/or rephrased sentences/ paragraphs are colored in red, just in order to be easy to be observed.
Round 2
Reviewer 1 Report
Comments and Suggestions for Authors
The manuscript needs to be thoroughly re-revised before being considered for publication. It still contains typos and sentences that are confusing. For example:
"In fact, generally, aging results also – of (paraphysiologic) various reasons – in a ”progressive chronic pro-inflammatory state”, and this is considered a basic bologic issue, encountered in different age-releted siccknesses, including AD"
"Some literature data upon related trials report that this type of intervention, target- ing specific acupoints, ”replenish qi [supporting the related basic energetic balance flow across the construct structure of acupuncture meridians – o. n.10, resolve phlegm, and promote blood circulation can reduce Alzheimer's Disease Assessment Scale-cognitive subscale (ADAS-cog) and Clinician's Interview-Based Impression of Change-Plus (CI- BIC-Plus) scores of AD patients” (77)."
2. I strongly recommend the authors to remove the quotations. Some of the sentences don't need to be within quotes.
For example: ”Taking into account the, fairly, always limited editorial space, we shall not detail related methodological aspects”(76) Also why is there a citation next to this sentence?
3. Minor issue :APP should be expanded.
Comments on the Quality of English Language
Not upto the mark for publication.
Author Response
Dear Peer-Reviewer,
Thank you for your new thorough comments and contributive suggestions for the improvement of our
manuscript. Accordingly, please find herein, below, our point-by point answers:
- We have thoroughly re-revised, with extensive editing – confirmed including by the Editors’ check done using the iThenticate soft – and typos eliminating, so we hope the text is now clear enough. Additionally, we have checked the sentence you have indicated to be ”confusing”, but we really do not understand what is confusing in an assertion quite largely encountered in the literature, which we have mentioned in the text, i.e. that aging consists including in the progression of a chronic pro-inflammatory state in the human old bodies ?:
In fact, generally, aging results also – of (para-physiologic) various reasons – in a ”progressive chronic pro-inflammatory state”, and this is considered a basic biologic issue (together with ”the critical role of a dysregulated immune system in promoting persistent neuroinflammation”) (206), encountered in different age-related sicknesses, including AD (38) and Parkinson's disease (PD) (206).
Regarding the sentence "Some literature data upon related trials report that this type of intervention, targeting specific acupoints, ”replenish qi [supporting the related basic energetic balance flow across the construct structure of acupuncture meridians – o. n.10, resolve phlegm, and promote blood circulation can reduce Alzheimer's Disease Assessment Scale-cognitive subscale (ADAS-cog) and Clinician's Interview-Based Impression of Change-Plus (CI- BIC-Plus) scores of AD patients” (77).": Resolved. Please see the related revised form of it (all the new added words and/or rephrased sentences/ paragraphs are written bolded and colored in brick-red/tile, just in order to be easier observed):
Some literature data upon related trials report that this type of intervention, targeting specific acupoints, especially which support the related basic energetic balance flow across the construct structure of acupuncture meridians9 ”replenish qi, resolve phlegm, and promote blood circulation”, outcomes objectified through favorably modified scores on the: Alzheimer's Disease Assessment Scale-cognitive subscale (ADAS-cog) and Clinician's Interview-Based Impression of Change-Plus (CIBIC-Plus) scores of AD patients (77)
- Hopefully Resolved: we re-emphasize we have thoroughly re-revised, with extensive editing of the text, and eliminated most of the quotes. There remained some because, as we have explained in our previous answer to you, as you could determine, not few works present different non-pharmacological
interventions quite enthusiastic, and given the fact that practically at present there is no cure for AD, and not even any decisively effective therapeutic-rehabilitative interventions able to provide spectacular
improvements (inclusively) regarding this disease, using quotations we considered to thus keep a
relative distance towards some too enthusiastic assertions.
Regarding the example you have mentioned: ”Taking into account the, fairly, always limited editorial space, we shall not detail related methodological aspects”(76) – Resolved. Please see the related revised form of it (all the new added words and/or rephrased sentences/ paragraphs are written bolded and colored in brick-red/tile, just in order to be easier observed):
To be also noted that as from objective editorial customs, including this paper must, normally, frame within a reasonably extension; so, herein ”we shall not detail related methodological aspects” (76)
We kept the quotes for the last few words of the above sentence as this is a formulation we have used also in other systematic literature reviews we have previously published internationally
- ”Minor issue :APP should be expanded.”
Kindly please: could you be more specific with this last requirement ? Thank you
Reviewer 2 Report
Comments and Suggestions for Authors
The revised version has improved significantly. I'm satisfied with the revised version for publication.
Comments on the Quality of English LanguageThe quality of English looks good to me.
Author Response
Thank you very much!
Round 3
Reviewer 1 Report
Comments and Suggestions for Authors
Please expand all the abbreviations used in the paper. Example : APP - Amyloid Precursor Protein.
I found a lot of minor typos and grammatical errors in the manuscript. These should be corrected. Otherwise, the manuscript is okay to be published.
Comments on the Quality of English LanguageThere are several typos and grammatical errors in the manuscript.
Author Response
Dear Peer-Reviewer,
Thank you for your new thorough comments and contributive suggestions for the further improvement of our manuscript and for giving to our paper the ”okay to be published”.
Accordingly, please find herein, below, our point-by point answers:
”Please expand all the abbreviations used in the paper. Example : APP - Amyloid Precursor Protein.
I found a lot of minor typos and grammatical errors in the manuscript. These should be corrected. Otherwise, the manuscript is okay to be published.” Resolved. Please see the related last revised form of our manuscript (all the new added words – abbreviations, too – and/or rephrased sentences/ paragraphs, including with supplementary typos corrections, are written colored in red, just in order to be easier observed).
However, like we have previously written, if still necessary – as we are not native English speakers – we shall invoke the IJMS professional editing service.
Once again: thank you !